# Internal Thread Defect Generation Algorithm and Detection System Based on Generative Adversarial Networks and You Only Look Once

**DOI:** 10.3390/s24175636

**Published:** 2024-08-30

**Authors:** Zhihao Jiang, Xiaohan Dou, Xiaolong Liu, Chengqi Xue, Anqi Wang, Gengpei Zhang

**Affiliations:** School of Electronic lnformation and Electrical Engineering, Yangtze University, Jingzhou 434100, China; jiangzhihao647@163.com (Z.J.); 13227676170@163.com (X.D.); 13349924158@163.com (X.L.); chengqixue2020@163.com (C.X.);

**Keywords:** internal thread, defect detection, machine vision, deep learning

## Abstract

In the field of industrial inspection, accurate detection of thread quality is crucial for ensuring mechanical performance. Existing machine-vision-based methods for internal thread defect detection often face challenges in efficient detection and sufficient model training samples due to the influence of mechanical geometric features. This paper introduces a novel image acquisition structure, proposes a data augmentation algorithm based on Generative Adversarial Networks (GANs) to effectively construct high-quality training sets, and employs a YOLO algorithm to achieve internal thread defect detection. Through multi-metric evaluation and comparison with external threads, high-similarity internal thread image generation is achieved. The detection accuracy for internal and external threads reached 94.27% and 93.92%, respectively, effectively detecting internal thread defects.

## 1. Introduction

In modern industrial manufacturing, threads, as critical components, are widely used in the aerospace, mechanical manufacturing, and instrument manufacturing fields [1], playing a crucial role in transmission, sealing, and connection. The quality of threads directly affects the service life, safety, and reliability of products, making efficient and accurate thread inspection particularly critical. However, internal thread defect [2] detection is challenging, which poses difficulties for the reliability and lifespan of products. Therefore, developing effective internal thread defect detection technology is of great significance for improving product quality. Although traditional image processing-based detection methods, including image segmentation [3], manual feature extraction [4], and defect recognition [5], have certain advantages in cases with limited annotated data, these methods gradually expose limitations when dealing with complex production environments as technology advances, making it difficult to meet the demands for automation and intelligence in modern industry. Therefore, exploring more advanced internal thread detection technologies has become an urgent priority in terms of enhancing detection efficiency and product quality.

With the rapid development of deep learning technology, surface defect detection methods based on deep learning have gradually gained widespread attention in both theory and practice. These methods can significantly improve the accuracy and robustness of defect detection when sufficient data are available. However, the scarcity of industrial defect samples poses a challenge to detection accuracy. To address this issue, researchers have proposed various few-shot learning methods, including metric learning [6], data augmentation [7], transfer learning [8], model fine-tuning [9], semi-supervised learning [10], weakly supervised learning [11], and unsupervised learning [12,13]. Metric learning mitigates overfitting by optimizing sample distances but performs poorly when data diversity is insufficient. Model fine-tuning and semi-supervised learning have potential in utilizing unlabeled data but show unstable results when data are extremely imbalanced. Weakly supervised and unsupervised learning reduces the dependency on precise annotations but usually requires complex algorithm designs to prevent learning incorrect patterns. In recent years, uncertainty perception metric networks (UPMNs) [14] have demonstrated excellent performance in addressing issues related to insufficient samples and data imbalance by introducing uncertainty quantification, allowing them to maintain high detection accuracy in environments with significant noise, ambiguous features, and high data complexity. The core design of a UPMN focuses on enhancing robustness through uncertainty quantification. Since internal thread defects typically exhibit clear and stable characteristics, the complexity and computational burden could lead to decreased efficiency. Additionally, internal thread detection relies on high-precision image acquisition systems, reducing the need for uncertainty quantification, which limits the advantages. In this context, a combination of defect detection architectures derived from data augmentation and transfer learning is more suitable. Data augmentation improves model generalization by expanding the dataset, while transfer learning-based defect detection provides efficient real-time detection capabilities, making them more advantageous in internal thread defect detection.

Traditional data augmentation techniques include image translation and flipping, color transformation, noise addition, blurring, affine transformation, and perspective transformation. However, these techniques are limited by their reliance on existing samples and strong dependency on features. Since GANs can address the shortcomings of traditional data augmentation in generalization, they have been widely applied in various fields, also extending to different variant forms. Kazeminia [15] pointed out that, although GANs perform well in tasks such as denoising, reconstruction, and segmentation of medical images, GANs face difficulties in handling all situations due to the high diversity and complexity of medical images. A GAN requires a large amount of high-quality labeled data for training, which is difficult to obtain in industrial applications, limiting its application in thread detection. Esfahani and Latifi [16] introduced five application areas of GAN technology but pointed out that a GAN’s training process is complex and unstable, being prone to mode collapse and gradient vanishing, especially when dealing with complex tasks, leading to instability in internal thread defect detection. If data are limited or computational resources are insufficient, the generative effect of a GAN cannot meet expectations, limiting the effectiveness of the detection model. Huang Z [17] proposed a new method, a DU-GAN, which improves the denoising quality of low-dose CT images in both image and gradient domains through a U-Net discriminator. However, DU-GAN lacks generalization ability, and, in terms of internal thread detection, if the dataset changes, consistent detection performance cannot be maintained. Ramwala O A [18] proposed a new method called Reminiscent Net, which utilizes the generative ability of a Conditional GAN (CGAN) to repair and reconstruct damaged ancient documents and photographs. However, when facing complex industrial detection, this specific application is difficult to effectively transfer to internal thread defect detection. Zhang [19] used a Defect-GAN to generate defects on normal surface images and restore normal images through the recovery process, achieving defect synthesis. Since thread defects have complex periodic structures, general defect generation models cannot easily capture and simulate microstructural details with sufficient precision. Duan [20] proposed the use of a Defect-aware Feature Generation GAN (DFMGAN), which captures and reflects the diversity and complexity of defects through defect-aware feature manipulation, generating realistic and diverse defect images with limited defect-free images and a small number of defect images, thereby providing more high-quality training data for model training and improving the accuracy and robustness of detection algorithms. Given the complex geometric surface of internal threads and the morphological characteristics of defect overlap, data augmentation technology needs to have defect-aware feature manipulation capabilities. Therefore, this paper selects a DFMGAN to support internal thread defect image data augmentation technology.

Defect detection is of great significance in the industrial field, and traditional defect detection methods, such as the threshold segmentation method [21], edge detection method [22], and template matching method [23], have shortcomings. Transfer learning methods have made certain improvements in detection speed and accuracy. However, a Fast Region-based Convolutional Network (Fast R-CNN) [24] is not fast enough because it requires two-stage processing, and a Single Shot Multibox Detector (SSD) [25] performs poorly in small object detection and has low accuracy in complex backgrounds. You Only Look Once (YOLO) [26], as an advanced object detection technology, has significantly improved detection accuracy and speed. The evolution of YOLO from the initial version to the latest version demonstrates significant technological progress. Terven et al. [27] conducted a systematic review detailing the innovations and contributions of each generation of YOLO algorithms. Li et al. [28] proposed YOLOv6, which surpasses other mainstream detectors in both average precision and frames per second, while its improved version YOLOv8 is optimized for aerial image recognition, showing outstanding performance and robustness. Reis et al. [29] explored the application of YOLOv8 in real-time flying object detection, emphasizing its ability to balance inference speed and average precision. Meanwhile, Su et al. [30] utilized YOLOv8 for traditional Chinese medicine slice detection, confirming its advantages in regard to high precision across various categories, efficient model size, and frames per second (fps). Overall, YOLOv8 adopts an advanced network architecture that further enhances model performance and flexibility, achieving an unprecedented balance between precision and speed, optimizing the training process to improve model training efficiency and result stability and enhancing the detection capability of small and dense targets. Considering the internal thread surface’s characteristics of scratches, broken profiles, corrosion, and other small and densely packed defects, YOLOv8 is undoubtedly the optimal choice for defect detection from the perspectives of speed, accuracy, and robustness.

Based on the importance of internal thread defect detection and the specificity of the challenges faced by detection technology, this paper proposes an internal thread defect detection method based on DFMGAN data augmentation and YOLOv8 defect detection. This paper will explain the implementation process and effectiveness of this method through steps such as internal thread image acquisition systems, image preprocessing, model training and Effectiveness Verification.

## 2. System Theory

In this paper, an innovative defect detection method based on an internal thread image acquisition device is proposed. The work presented in this paper is divided into two main parts, hardware and software, as illustrated in Figure 1.

The hardware part focuses on utilizing a computer-controlled robotic arm to capture images, while the software part processes the captured images. To address the need for internal thread defect detection, this paper proposes a high-precision detection method based on a combination of robotic arm control and fisheye lenses. The entire process consists of the following four key steps: data acquisition and preprocessing, DFMGAN training, YOLOv8 training, and defect detection.

Firstly, this paper designs a robotic arm control system equipped with a fisheye lens to achieve efficient image capturing throughout the detection process. The robotic arm is controlled via a computer, ensuring that the fisheye camera can penetrate into the internal threads to capture comprehensive images. The advantage of the fisheye camera lies in its wide field of view, which can cover the entire area of the internal threads in a single shot, thereby avoiding the cumbersome steps of multiple shootings and improving detection efficiency.

Due to the unique optical structure of the fisheye lens, the captured images often exhibit significant distortion, affecting the accuracy of subsequent image processing and defect detection. Therefore, after acquiring the images, this paper first performs distortion correction. By utilizing precise distortion correction algorithms, the fisheye images are restored to resemble the actual objects, providing an accurate data foundation for the subsequent processing steps.

The corrected images undergo a two-step preprocessing process that includes filtering and enhancement. Various filtering techniques are employed to remove noise from the images, and the filtered images are then enhanced to highlight the features of the internal thread defect areas. Prior to defect detection, to improve the model’s generalization ability and robustness, this paper performs data augmentation on the preprocessed images.

Finally, this paper adopts a general defect detection algorithm for internal thread defect detection. By training and optimizing the defect detection algorithm, this paper achieves high-precision detection of internal thread defects.

### 2.1. Image Acquisition Equipment

This paper designs a more efficient internal thread defect detection system [31], as shown in Figure 2. The system comprises a framework, robotic arm, a Light Emitting Diode (LED) light strip, and fisheye cameras. The robotic arm is connected to the camera fixture, with an LED light strip and fisheye camera being mounted on the surface of the connecting rod. Due to the small inner diameter of some threaded pipes, the camera’s field of view within the pipes may lack overlapping regions, resulting in missing information.

The vision system consists of six 1680 × 1080 fisheye cameras, arranged in two groups of three cameras each. Each group is arranged in a 120° circular configuration, with the two groups positioned front-to-back and offset by 60° to ensure the entire vision system comprehensively covers the internal threads. Compared to controlling the vision system via a stepper motor, replacing it with a robotic arm drive enhances the system’s adaptability to various types of threaded pipes while meeting the image acquisition requirements.

Furthermore, the acquisition method is optimized, as illustrated in Figure 3. The fisheye cameras remain continuously active, and the robotic arm, controlled via a computer, extends the camera into the internal threads for imaging. High-quality images are extracted from the captured video through a well-designed frame extraction rate. Considering the video frame rate is 30 fps, this article extracts one frame every six frames, resulting in five fps. This frame extraction rate balances image detail capture and processing efficiency, ensuring sufficient detail while generating a manageable number of images for subsequent analysis.

After the acquisition of an internal thread image, it is necessary to further preprocess the acquired image to obtain high-quality image data.

### 2.2. Image Preprocessing

After image acquisition, a series of image processing steps are required to ensure the accuracy and reliability of defect detection. The acquired image is preprocessed, which involves distortion correction, denoising, image enhancement, and other operations to improve the quality and clarity of the image.

The image correction process involves selecting an appropriate vision system to characterize geometric distortions, estimating the parameters describing image formation, and ultimately performing distortion correction based on the camera model. The choice of the vision system typically depends on the lens type, as standard perspective lenses are suitable for imaging small areas and small objects while fisheye lenses are ideal for capturing wide-angle views and large objects, thereby acquiring more information.

In the image distortion correction process, this paper calibrates the camera using a chessboard pattern and employs a correction method based on the fisheye camera model to correct the distortion in the captured images, providing a foundational basis for subsequent image processing. Lee [32] proposed a fisheye lens radial distortion correction algorithm utilizing linear characteristics which performs well in terms of stability and distortion error but may increase computational complexity and time costs due to its complex parameter search and evaluation steps. Park [33] introduced an image enhancement method for fisheye cameras which successfully removes geometric distortion from fisheye lenses but may introduce jagged edges and blur artifacts during the correction process, requiring further processing. Chen and Han [34] provided a near-distance fisheye camera model that reflects the relationship between imaging distortion and object distance that is potentially more suitable for close-range shooting scenarios and less effective in long-distance or complex scenes.

During the acquisition of thread images, limitations of the imaging system, information transmission, and storage devices may introduce noise interference, resulting in irrelevant noise information in the local thread images. This noise can affect subsequent feature point extraction and registration, reducing accuracy and increasing computational load. To address this issue, it is necessary to perform filtering on the local thread images to reduce the impact of noise on registration accuracy. Common image filtering methods include mean filtering [35], median filtering [36], Gaussian filtering [37], and bilateral filtering [38].

Although these traditional filtering methods can reduce noise to some extent, they have some common drawbacks. Firstly, these methods usually rely on fixed filtering kernels or windows which cannot adapt to complex noise patterns, often leading to loss of image details or blurring of edges. Secondly, traditional filtering methods have limited effectiveness in handling high-density noise, which may introduce artifacts or retain some noise. Additionally, the effectiveness of these methods highly depends on parameter settings; improper adjustment can significantly affect the filtering results and make it difficult to maintain consistency across different images and noise conditions.

In contrast, deep learning-based filtering methods have significant advantages. Firstly, deep learning models can automatically learn complex features in images, effectively separating noise from useful signals. Secondly, deep learning methods have high robustness and generalization capabilities, enabling stable performance under various complex imaging conditions. Moreover, by using pre-trained models and transfer learning, efficient training can be achieved on relatively small datasets, further improving filtering effectiveness and processing speed. A Denoising Convolutional Neural Network (DnCNN) [39] is a convolutional neural network specifically designed for denoising tasks. The network takes as input a noisy image of size [256 × 256 × 3] and outputs a denoised image of the same size. A DnCNN is composed of 17 convolutional layers. The first layer uses 64 convolutional filters of size [3 × 3 × 3], layers 2 through 16 each use 64 filters of size [3 × 3 × 64], and the 17th layer uses a single filter of size [3 × 3 × 64]. Each convolutional layer is followed by a batch normalization layer, and ReLU activation functions are applied after the first 16 convolutional layers to enhance the network’s stability and nonlinear expression capability. During training, the DnCNN optimizes its denoising performance by minimizing the mean squared error (MSE) between the denoised image and the original noise-free image.

In this work, we use Structural Similarity Indexes (SSIMs) and Peak Signal-to-Noise Ratios (PSNRs) to evaluate the filtering effect.

An SSIM is a measure of the similarity between two images. The specific formula for an SSIM is as follows, where $\mu_{x}$ and $\mu_{y}$ are the average values of images x and y, $\mu_{x}^{2}$ and $\mu_{y}^{2}$ are the variances of images x and y, $\sigma_{xy}$ is the covariance of images x and y, and $C_{1}$ and $C_{2}$ are small constants to stabilize the denominator.
(1)$$SSIM\left(x,y\right)=\frac{\left(2\mu_{x}\mu_{y}+C_{1}\right)\left(2\sigma_{xy}+C_{2}\right)}{\left(\mu_{x}^{2}+\mu_{y}^{2}+C_{1}\right)\left(\sigma_{x}^{2}+\sigma_{y}^{2}+C_{2}\right)}$$

A PSNR reflects the ratio between the maximum possible value of a signal and the power of distorting noise that affects the quality of its representation. The larger the value, the more similar the two images are and the better the quality. The specific formula is as follows, where MAX is the maximum possible pixel value of the image and MSE is the mean squared error between the images, defined as:(2)$$PSNR=10\log_{10}(\frac{MAX^{2}}{MSE})$$
(3)$$MSE=\frac{1}{mn}\sum_{i=0}^{m-1}\sum_{j=0}^{n-1}{\left[I\left(i,j\right)-K\left(i,j\right)\right]}^{2}$$

To enhance the quality of thread images, increase visual details, and improve their appearance, making them clearer, more vivid, and easier to analyze to better reveal texture details and defects, traditional image enhancement techniques play a vital role. Among them, the Retinex [40] algorithm is primarily used to adjust image brightness and color by simulating human eye perception of scenes, effectively improving the dynamic range and color reproduction of images. Histogram equalization [41] enhances contrast by optimizing the grayscale distribution of images, making it suitable for improving overall brightness and clarity. Adaptive histogram equalization [42] further enhances this by processing local areas of the image and more finely adjusting local contrast, which is especially suitable for scenes with strong brightness and darkness contrasts.

### 2.3. Image Data Generation

During the pre-training phase with defect-free images, we use the StyleGAN2 generative network to train on hundreds of defect-free images. By randomly sampling latent codes, we generate a variety of defect-free images. StyleGAN2 includes a mapping network and a synthesis network, where the former maps random noise to the latent space and the latter generates the final images through convolution. After training, the generative network encodes rich object features. In the transition to defect images, we attach defect-aware residual blocks and a defect-mapping network to the pre-trained generative network and train these additional modules on a small number of defect images. The defect-aware residual blocks are used to generate and manipulate defect regions, while the defect-mapping network outputs modulation weights by randomly sampling defect codes which modulate the weights of the residual blocks. The residual blocks extract features from the main feature maps to generate defect features and defect masks. The mode-seeking loss improves the diversity of generated defect images by minimizing the similarities between defect masks generated by different defect codes. Finally, a standard discriminator and a defect-matching discriminator are used to judge the authenticity of the images and the consistency between the images and defect masks, as shown in Figure 4.

Figure 5 shows the structure of the generator, consisting of a backbone (StyleGAN2) and a defect-aware residual block. The StyleGAN2 backbone consists of a mapping network and a composite network, where the mapping network maps a randomly sampled object code ($Z_{object}$) to the feature space and outputs $W_{object}$. The synthetic network starts with the initial feature map (c) and gradually generates the final image through a series of convolutional composite blocks; the weight of each composite block is modulated by wobject, and the synthetic network finally converts the feature map to an RGB image via the ToRGB module. The defect-sensing residuals are attached to StyleGAN2 at specific resolutions (e.g., 64 × 64) to generate defect masks and manipulate feature maps. The ToMask module is similar to the ToRGB module but outputs a single-channel defect mask (M). The residuals block adds Fdefect features to the defect area based on the defect mask, leaving the non-defect area unchanged. The defect-mapping network maps the generated defect code ($Z_{defect}$) to a modulation weight ($W_{defect}$), which is used to modulate the weight of the defect-sensing residual block.

Figure 6 shows the structure of the discriminator. A DFMGAN uses two discriminators to provide a supervisory signal. First, the StyleGAN2 discriminator is responsible for determining the authenticity of the generated image, and its structure is similar to that of the original StyleGAN2 discriminator. Secondly, the defect-matching discriminator is used to judge the consistency of the generated image and the defect mask. The discriminator inputs the Mosaic image and mask and outputs the matching judgment. Compared with a StyleGAN2 discriminator, the defect-matching discriminator has fewer convolutional layer channels to meet the training requirements of a small number of defect images.

The mode-seeking loss uses two random defect codes, $Z_{defect}^{1}$ and $Z_{defect}^{2}$. The defect-mapping network outputs two corresponding modulation wexights, $W_{defect}^{1}$ and $W_{defect}^{2}$. The entire model uses $W_{defect}^{1}$ and $W_{defect}^{2}$ along with the same $W_{object}$, to generate defect masks $M^{1}$ and $M^{2}$. Then, the DFMGAN minimizes the mode-seeking loss, as shown in Formula (4).
(4)$$L_{ms}=\frac{\|W_{defect}^{1}-W_{defect}^{2}{\|}_{1}}{\|M^{1}-M^{2}{\|}_{1}}$$

The goal of the mode-seeking loss is to maximize the differences between defect masks generated by different $W_{defect}$. Specifically, when a wdefect is different, the resulting defect masks $M^{1}$ and $M^{2}$ should exhibit significant variability. In other words, the DFMGAN aims to ensure that, for different defect input codes, the generated defect masks display notable visual distinctions. This enhances the model’s ability to differentiate between various defect features, thereby improving its overall performance in distinguishing different defect characteristics.
(5)$$L\left(G,D,D_{match}\right)=L_{StyleGAN}\left(G,D\right)+L_{match}\left(G,D_{match}\right)+\lambda L_{ms}\left(G\right)$$

Formula (5) is the overall loss function of the DFMGAN model.

The total loss function $L\left(G,D,D_{match}\right)$ of the DFMGAN model is used to optimize the generator G, the discriminator D, and $D_{match}$. $L_{match}\left(G,D_{match}\right)$ is the original loss function used by StyleGAN2, which optimizes the generator and discriminator through adversarial loss, path length regularization, and R1 regularization. $L_{\mathrm{StyleGAN}}\left(G,D\right)$ is the loss function associated with the discriminator $D_{match}$. $D_{match}$ is used to assess how well the generated and real images match on specific attributes. $\lambda L_{ms}\left(G\right)$ is a mode-seeking loss. The mode-seeking loss $L_{ms}$ has a weight parameter λ, which is a hyperparameter used to adjust the weight of multi-scale losses in the total loss. The goal of mode-seeking loss is to maximize the differences between defect masks generated by different $W_{defects}$.

This loss function combines the basic loss, matching loss, and pattern seeking loss of StyleGAN2 and realizes effective distinction of different defect features and high-quality image generation by alternating optimization of the generator and discriminator.

After completing image generation, to rigorously and accurately evaluate the generated images and the model, this paper conducts a qualitative analysis of various evaluation metrics (SSIM [43], UPMNs [44], Weighting Similarity–Manhattan Distances (WSMDs) [45], and PSNRs [46]).

The Weighting Similarity–Manhattan Distance combines weighting similarity and the Manhattan distance, aiming to evaluate the differences in different parts of the image. In this method, weights are assigned to different regions based on their importance, so that certain important regions receive more attention when calculating differences. The specific formula is as follows:(6)$$D_{WSMD}\left(I_{1},I_{2}\right)=\underset{i}{\sum }w_{i}\cdot \left|x_{i}-y_{i}\right|$$
where $D_{WSMD}$ represents the Weighting Similarity–Manhattan Distance, x and y are two images, $w_{i}$ is the weight of the i-th pixel, and $\left|x_{i}-y_{i}\right|$ is the absolute difference of the i-th pixel.

Uncertainty scoring is used to measure the reliability of model predictions. A common method is based on entropy calculation. The higher the entropy, the greater the uncertainty in the model’s predictions. Uncertainty scoring can be used to assess the confidence in generated images. For classification tasks, given the prediction probabilities of a sample, $p_{i}$ being the probability of the i-th class, entropy is defined as follows:(7)$$H\left(P\right)=-\sum_{i=1}^{k}p_{i}log\left(p_{i}\right)$$

A comprehensive evaluation of each metric is performed and tabulated, as shown in Table 1.

In summary, when evaluating image quality, the combination of an SSIM and the uncertainty perception metric network provides a comprehensive assessment of the reliability of model predictions. This approach not only evaluates the perceptual quality of the image but also assesses the model’s confidence through uncertainty evaluation, offering a more complete detection solution. Therefore, the evaluation metrics for images generated by the DFMGAN are selected as the SSIM and the uncertainty perception metric network (to distinguish from the filtering evaluation metrics, use “FILTER” and “GAN” labels when creating tables).

### 2.4. Defect Detection

You Only Look Once version 8 (YOLOv8) is an advanced object detection algorithm characterized by predicting the position and category of objects simultaneously during a single forward pass, significantly improving detection speed and accuracy. The specific model structure is as follows:

YOLOv8 employs an improved Backbone and Neck network structure. The Backbone network is used to extract image features, while the Neck network further processes these features to generate the final detection results.

The Backbone network typically uses CSPDarknet53, which combines the advantages of a Cross Stage Partial Network (CSPNet) and Darknet53. A CSPNet effectively reduces the model parameters and computational complexity while maintaining high accuracy. The formula is represented as follows:(8)CSPBlock(X)=X+F(X)
where, F(X) represents the feature map after passing through the convolutional layer, normalization layer, and activation layer.

The Neck network uses the combination of a Path Aggregation Network (PANet) and a Feature Pyramid Network (FPN), which can better fuse features of different scales and improve the ability to detect small objects.
(9)$$P_{i}=FPN\left(C_{i}\right)+PANet\left(C_{i}\right)$$
where, $C_{i}$ denotes the feature maps of different levels and $P_{i}$ denotes the fused feature maps.

The detection head of YOLOv8 uses the Anchor Box mechanism and combines Focal Loss and Complete IoU Loss (CIoU Loss) to improve the detection accuracy and convergence speed.

YOLOv8 adopts Focal Loss to deal with class imbalance.

CIoU Loss is used to regress bounding boxes, taking into account IoU, center point distance, and aspect ratio, and the formula is expressed as follows:(10)$$CIoU=IoU-\frac{\rho^{2}\left(b,b^{*}\right)}{c^{2}}-\alpha \frac{v}{1-IoU+v}$$
where, ρ denotes the distance between the center points of the bounding box, c denotes the diagonal distance of the minimum closure region, and α and v are the conditioning parameters.

YOLOv8 optimizes the training strategy in many ways, including data augmentation, label smoothing, and the balance of positive and negative samples.

The Mosaic data augmentation method is used to concatenate four images into one, which increases the diversity and robustness of the training data. The formula is as follows:(11)$$Mosaic\left(X_{1},X_{2},X_{3},X_{4},\ldots ,X_{i}\right)=Concat\left(X_{1},X_{2},X_{3},X_{4},\ldots ,X_{i}\right)$$
where, $X_{i}$ denotes the original image.

Label smoothing is used to mitigate the overfitting problem by reducing the true label value slightly to reduce the confidence of the model.

In summary, YOLOv8 has significant advantages in industrial detection, and its high-precision algorithm ensures the accurate detection of defects and anomalies. Its high-speed processing ability meets the real-time detection needs of industrial production lines. At the same time, it has wide applicability and can deal with a variety of materials and complex backgrounds. Its strong generalization ability still performs well in diverse industrial environments, ensuring efficient performance in different datasets.

## 3. Experiment

A YOLOv8 quantitative experiment is used to judge the applicability of the generated image based on a DFMGAN. The specific flow chart is shown in Figure 7, where real external thread (RET) is the real external thread defect map, generated external thread (GET) is the generated external thread image defect map, and generated internal thread (GIT) is the generated internal thread defect map. By using a DFMGAN to generate images based on real internal and external threads, and comparing the training results of real external thread images with the training results of generated external thread images based on YOLOv8, the effectiveness of the images generated by a DFMGAN is verified. Then, the generated internal thread images are trained based on YOLOv8 and the results are compared with the generated external thread training results. The validity of the generated internal thread image dataset is proved.

In the experiment of internal thread defect detection, a 100 mm diameter internal thread is selected as the experimental object to ensure detection accuracy and reliability. The specific parameters are as follows: a nominal diameter of 100 mm, a pitch of 2 mm, a thread angle of 60 degrees (metric thread), a tolerance grade of 7H, a thread length of 50 mm, and the thread type is coarse thread. The reasons for choosing these parameters are as follows: the 100 mm diameter thread is large, making it suitable for resolution testing of defect detection equipment; the pitch of 2 mm and the thread angle of 60 degrees comply with common metric standards, facilitating comparative experiments under different equipment and conditions; the 7H tolerance grade ensures manufacturing accuracy while providing appropriate detection challenges; and the 50 mm thread length ensures a sufficient detection range that covers multiple parts of the thread, facilitating a comprehensive assessment of the detection system’s performance. Through this experimental template, the effectiveness of internal thread defect detection methods can be systematically evaluated, providing reliable data support for subsequent technical optimization and practical applications.

### 3.1. Data Acquisition and Image Preprocessing

Based on the analyzed models, this paper constructs the vision system, as shown in Figure 8.

Cameras are used to capture images and obtain corner coordinates by detecting the chessboard corners, followed by subpixel optimization of these corner coordinates. By referencing the principles of subpixel optimization, the camera’s intrinsic parameters are estimated and calculated. Following Zhang’s calibration method, initial calibration results are obtained, and the intrinsic parameters and distortion vectors are continuously optimized based on new data, calculating the mapping matrix to achieve distortion-free and corrected transformation relationships.

Upon completing the camera calibration, we use the obtained intrinsic parameter matrix and distortion parameter matrix to perform image distortion correction through appropriate matrix operations. This process ensures accurate correction of the viewing angles, enhancing image quality and analysis accuracy. The results are shown in Figure 9.

In this study, images with a noise level of 15 dB were processed using mean filtering, median filtering, non-local means filtering, 3D block matching filtering, and Denoising Convolutional Neural Network (DnCNN) filtering for comparison. The pairs of various filtering algorithms are shown in Table 2.

In this study, PSNRs and SSIMs were used as image quality evaluation metrics. According to the analysis in the table, the image filtering effects of 3D block matching filtering and the DnCNN are better than those of mean filtering, median filtering, and non-local means filtering. However, the DnCNN has a faster computation speed, making it suitable for real-time defect detection in internal threads. Therefore, this project chooses DnCNN-based filtering.

Under low-light conditions, images often suffer from low visibility. To address this issue, the illumination conditions within the image are precisely modeled and estimated, followed by appropriate enhancement based on these estimates. This approach can significantly improve the brightness and clarity of the images. The contrast of the enhanced images is shown in Figure 10, comparing enhancements using Single-Scale Retinex (SSR), Multi-Scale Retinex (MSR), and Multi-Scale Retinex with Color Restoration (MSRCR) algorithms. These techniques draw inspiration from the way the human eye perceives light and color, effectively improving the visual quality of images captured under suboptimal lighting conditions. Among them, the Multi-Scale Retinex with Color Restoration (MSRCR) method demonstrated the best performance and, therefore, was chosen for image enhancement.

After image preprocessing, representative frames are selected from the video stream, redundant information is removed, key image information is retained, and these key frames are spliced to generate a continuous and high-quality image sequence. The specific effect is shown in Figure 11.

### 3.2. Image Data Generation Based on DFMGAN

The DFMGAN model consists of a generator and two discriminators. The generator takes a latent vector of size 128 as input and outputs a defect image with the dimensions [256 × 256 × 3]. This generator is built on the StyleGAN2 architecture, with the model being enhanced by adding defect-aware residual blocks at the following different resolutions: [64 × 64], [128 × 128], and [256 × 256]. These residual blocks introduce specific defect features into the generated images, modulated by a defect-specific latent code of size 128, ensuring that the generated images realistically represent the required defect characteristics.

In the implementation, the generator is constructed using the following key modules: First, the input latent vector passes through a Mapping Network, which maps the input vector into an intermediate latent space. In a DFMGAN, this mapping process involves multiple fully connected layers, each followed by an activation function. The mapped vector is then fed into the Synthesis Network, which generates the final defect image through a series of convolutional layers, feature map size adjustments (such as upsampling), and defect-aware residual blocks. During the generation process, the generator gradually increases the image resolution as needed and inserts defect-aware residual blocks at the [64 × 64], [128 × 128], and [256 × 256] resolutions to ensure that the defect features in the images are fully expressed.

The two discriminators in the DFMGAN model have distinct tasks. The main discriminator follows the StyleGAN2 structure, receiving an input image of size [256 × 256 × 3] and determining its authenticity. This discriminator extracts image features through multiple convolutional layers and, in the final layer, outputs the probability that the image is real. The defect-matching discriminator is specifically designed to assess the matching of defect areas in the generated images. It takes defect masks and the corresponding parts of the generated image as inputs, compares these at a resolution of [64 × 64], and outputs a matching score. The defect-matching discriminator uses adjusted convolution kernels and pooling operations to accurately identify small defects in the images.

The model is trained using the Adam optimizer, with the learning rate for both the generator and discriminators set to 0.001. During training, the model also employs various data augmentation techniques to improve the diversity and robustness of the generated images. Through the aforementioned design and training strategies, the DFMGAN model is capable of maintaining high detail in the generated images while ensuring computational efficiency, and it demonstrates a strong capability in regard to generating high-quality images that contain realistic defects.

In this paper, based on the above DFMGAN theory, generated internal and external thread images are generated, as shown in Figure 12. The left picture is the generated external thread image and the right picture is the real external thread image. In Figure 13, the left picture is the generated internal thread image and the right picture is the real internal thread image. The performance metrics for external and internal thread defect detection using DFMGAN-based models are shown in Table 3.

The arrows in the table indicate the significance of the metrics: the upward arrow next to SSIM (GAN) suggests that a higher value is better, as it reflects greater similarity to the reference image, indicating better image quality. Meanwhile, the downward arrow next to Uncertainty Score (GAN) indicates that a lower value is preferable, as it signifies greater stability and reliability in the generated image. In essence, the upward arrow means “the higher, the better”, while the downward arrow means “the lower, the better”. 

SSIM (GAN) values of 0.7933 and 0.7371 are relatively high, demonstrating a high similarity between the generated and real images in these aspects. The SSIM (GAN) values range from 0 to 1, with higher values indicating greater image similarities.

Uncertainty scores of 0.3986 and 0.4031 indicate that the model has a high level of confidence in the generated images, suggesting that these images are very close to real images in certain aspects. Lower uncertainty scores generally imply a higher degree of feature matching between the generated and real images, indicating that the model perceives minimal differences between them.

For the generated images, they retain the defects of the real images and exhibit a high degree of similarity in the style of the threaded features, indicating that the DFMGAN model has a strong image generation capability.

### 3.3. Defect Detection Based on YOLOv8

Based on YOLOv8 quantitative experiments, the applicability of the DFMGAN model to generate images was judged. The process consists of the following steps:
Use the DFMGAN to generate internal and external thread defect images.YOLOv8 is used to train and compare the real external thread images and the generated external thread images to verify the effectiveness of the generated images.The generated internal thread images are trained based on YOLOv8 and compared with the training results of the generated external thread images to prove the effectiveness of the generated internal thread image dataset.

The YOLOv8 model receives an input image of size [256 × 256 × 3], directly generated by the DFMGAN, and uses CSPDarknet53 as the backbone network for feature extraction. CSPDarknet53 incorporates the Cross Stage Partial Network’s (CSPNet) design, which partially separates different stages of the convolutional network. This design effectively reduces the flow of redundant gradient information, lowering computational load while maintaining high accuracy. This is particularly crucial when detecting small defects, such as internal thread flaws. The backbone network progressively downsizes the input image and outputs a feature map of size [20 × 20 × 256].

The Neck component combines PANet and FPN architectures to generate multi-scale feature maps, enhancing the model’s ability to detect objects of various sizes. Specifically, the Neck outputs feature maps of sizes [20 × 20 × 512], [40 × 40 × 256], and [80 × 80 × 128]. These feature maps are fused with multi-scale information, improving the model’s performance in detecting objects across multiple scales. The FPN generates multi-scale features through a top-down pathway with lateral connections, allowing the model to capture more detailed information about small objects. The PANet further strengthens the flow of information in the downward path, enabling feature maps of different scales to complement each other during detection, thereby improving the model’s localization and classification accuracy.

The detection head of YOLOv8 is responsible for generating the final bounding boxes and class probabilities, outputting a tensor of size [N, 4 + C + 1], where N represents the number of detected objects, 4 represents the bounding box coordinates, C represents the number of classes, and the last dimension represents the confidence score. By using the CIoU loss function for bounding box regression, the model can more accurately predict the bounding boxes of objects. Focal Loss effectively addresses the issue of class imbalance, improving detection performance for small object classes.

During the training process, the YOLOv8 model employs the Adam optimizer with a learning rate set to 0.01, ensuring a balance between convergence speed and performance. To further enhance the model’s robustness, data augmentation techniques such as Mosaic and random erasing are introduced during training. These techniques help the model maintain stable detection performance even in complex and variable real-world environments.

## 4. Experiment Results and Analysis

### 4.1. Verification of Data Generation Effectiveness

Due to the inability to obtain sufficient defect images, forming an adequate training set for defect detection is challenging. To address this, this study uses virtually generated defect images as the training set to test their effectiveness in detecting real defects. Since there are not enough real defect images of internal threads available to compare the detection results of real and virtual training sets, this study selects external thread defects and generates virtual images using the aforementioned method for comparative experiments with real datasets.

Based on YOLOv8, the experimental environment uses Python language and the Tensorflow framework, and the Graphics Processing Unit (GPU) is a Nvidia RTX 4070 SUPER graphics card. The generated internal and external threads are quantitatively analyzed for defect detection.

The task is to use the YOLOv8 model for defect detection; the training will be conducted for 300 cycles with batch sizes of 16 and image sizes of 640. The model employs pre-trained weights and uses an automatically selected optimizer. The initial learning rate is 0.01, the learning rate decline rate is also 0.01, the momentum is set to 0.937, and the weight decay is 0.0005. For data augmentation, horizontal flipping, Mosaic augmentation, and random erasure are applied, while RandAugment is used for automatic augmentation. The validation set evaluates it and the label and confidence will be displayed. The overall configuration focuses on improving the training efficiency and model robustness to ensure that the model gradually converges to the optimal state.

Precision indicates the proportion of True Positive samples among all samples predicted as positive by the model. True Positive (TP) is the number of samples correctly predicted as positive by the model. False Positive (FP) is the number of samples incorrectly predicted as positive by the model. The formula is as follows:(12)$$Precision=\frac{TP}{TP+FP}$$

The level of precision reflects the accuracy of the model in predicting positive samples, i.e., how many of the samples predicted as positive by the model are truly positive.

Recall is a metric that measures the detection capability of the model, representing the proportion of actual positive samples correctly predicted as positive by the model. False Negative (FN): The number of samples incorrectly predicted as negative by the model. The formula is as follows:(13)$$Recall=\frac{TP}{TP+FN}$$

A high recall indicates that the model can identify most of the positive samples, i.e., the miss rate is low.

Average Precision (AP) measures the overall performance of the model at different recall thresholds.
(14)$$AP=\int_{0}^{1}Precision\left(Recall\right)d\left(Recall\right)$$

AP represents the average precision of the model at various recall levels, providing an evaluation of the combined performance of precision and recall.

Mean Average Precision (mAP) is the average of APs for all categories, used to measure the overall detection performance of the model across all categories.

N represents the total number of categories, and $AP_{i}$ is the average precision of the i-th category.
(15)$$mAP=\frac{1}{N}\sum_{i=1}^{N}AP_{i}$$

MAP is the average of all category APs, reflecting the comprehensive performance of the model in handling multi-category detection tasks. In practical applications, mAP is often set with different Intersection over Union (IoU) thresholds.

These metrics can comprehensively evaluate the detection performance of the model, including prediction accuracy, detection comprehensiveness, and performance at different thresholds. These metrics are widely used in the field of object detection to compare and optimize the performance of different models.

Based on the YOLOV8 model, the generated images and real images are, respectively, used as the training set, and the same set of real images is used as the validation set for detection and analysis. The quantitative pairs are shown in Table 4.

According to Table 2, the recall rate of the external thread generated images is lower than the recall rates of the real images, which are 81.31% and 90.70%, respectively. The reason is that the generated images cannot represent the complexity and diversity of all real scenes, and the performance of the model is worse than that of the model trained on real images when recognizing some targets in real images. However, the detection results of the generated images are more accurate, and the False Positive cases are significantly reduced, thus effectively improving the detection accuracy. Therefore, it is feasible to use a DFMGAN for dataset augmentation in the training of thread defect detection.

Some real external thread defect images are detected, as shown in Figure 14. The red box marks the defect on the external thread. In this paper, the public dataset defect is named scratch and the latter is the probability of identifying it as a defect, so the detection is more accurate. In the following figure, nine external thread images are randomly detected, and a total of 11 defects are found, among which eight defects are completely detected. A total of 10 correct defects are detected, and the average sampling detection rate is 90.91%.

### 4.2. Effectiveness Verification of Internal Thread Defect Detection

Due to the variety of internal thread defects and the difficulty of image acquisition, which leads to the lack of a defect training set, this paper uses a DFMGAN to generate internal thread defect images and trains them based on the YOLOv8 model to detect real defects in internal threads. By employing this method, the issue of insufficient data can be effectively addressed, significantly enhancing the model’s ability to detect internal thread defects. The specific training results are shown in Table 5, including standard deviations, which demonstrate the model’s stability and performance across different experimental conditions (SD stands for standard deviation).

The internal thread defect detection model based on YOLOv8 performs well in accuracy and mAP, reaching 88.25% and 84.68%, respectively. Compared with the results of external thread training, due to the complex structure of the internal thread, the accuracy is slightly lower but the recall rate and mAP are better. The overall performance is similar, with no significant model deterioration being observed, and an effective training dataset can be constructed for the internal thread defect detection model.

For the generated internal thread images, the process curve based on YOLOv8 multiple training is shown in Figure 15. These subfigures show the change trend in each loss index and evaluation index, respectively, and the model gradually converges and optimizes.

Part of the thread defect images were collected for detection, as shown in Figure 16. The red box marks the defect on the internal thread, which is named broken in this project. The latter is the probability of identifying it as a defect, and the detection is more accurate. In the following figure, eight internal thread images are randomly detected, and a total of 21 defects are found, among which 7 defects are completely detected. A total of 20 correct defects are detected, and the average sampling detection rate is 95.24%.

### 4.3. Discussion and Analysis

In this paper, compared with the use of a stepper motor to control the image acquisition system, by taking the measurement distance of 10 cm threaded pipe as an example, with an image acquisition range of 2 cm and a 10% image overlap area, the two groups of cameras reciprocate according to the motion strategy, stop shooting for 0.5 s, and, considering the rotation rate and deceleration gear ratio of the motor, the use time of 10 cm measurement distance is expected to be 27 s. The axial motion rate set by the manipulator is 13.3 mm/s, and the coverage time of the 10 cm thread detection area is about 15 s. According to the time comparison, the efficiency of the acquisition process is improved by about 44.4%, and the real-time performance of the internal thread defect detection is greatly improved. Compared with the traditional stepper motor, the use of a mechanical arm can accurately locate the thread axis, the acquisition process is more stable, and it is suitable for various specifications of threads.

Comparing the training results of real and generated external threads, the accuracy of training based on real external threads is 92.03%, while the accuracy of training based on generated external threads is 93.92%, which confirms the effectiveness of image generation based on the DFMGAN and can be used as a training dataset for defect detection.

Compared with the corrosion defect trained by Xiaohan Dou [31], the accuracy is 88.36% and the accuracy of the internal thread defect broken trained on the generated internal thread dataset in this paper is 89.54%. Therefore, the reliability of the training set generated based on the DFMGAN is further demonstrated.

Through the detection of real and generated defect images of external threads, this paper verifies the rationality of generating images based on a DFMGAN for training and successfully solves the problem of it being difficult to train due to the insufficient training set.

In addition, this system has strong portability. Based on the image acquisition system, it can be collected on more objects, such as pipelines and internal combustion engines, and when the image dataset is scarce, the image generation based on a GAN and the internal thread defect detection technology based on YOLO show excellent portability in the software, making it possible for the software to run across platforms and adapt to different hardware configurations. It provides strong support for defect detection in more fields.

In certain scenarios, the effect of the generated image may not be satisfactory. This indicates that the model still has some limitations when dealing with these special cases. Deep learning models, especially GANs, are often considered as “black box” models that struggle to mechanize. This may affect its practical applications in medicine and other demanding fields.

Improvements can be made in the following areas, specifically preparing and training the dataset for different scenarios, such as training specifically on images in low light conditions to improve the performance of the model in these scenarios. Combined with the multi-task learning framework, the model is allowed to deal with multiple tasks at the same time in the training process to improve its adaptability to different scenarios. The user feedback mechanism is introduced to continuously optimize the model according to the user’s feedback to improve its practical application effect and credibility.

## 5. Conclusions

Based on multi-camera vision technology, this paper addresses the efficiency and accuracy issues of internal thread defect detection through software and hardware design and selection. The main contributions are as follows:
The performance improvement of the internal thread image acquisition system is achieved, while the video frame extraction technology optimizes the image acquisition efficiency.The training samples generation of internal thread is proposed based on a DFMGAN, addressing the shortage of high-quality training samples required for internal thread defect detection.The model of internal thread defect detection based on YOLOv8 is established, achieving high accuracy detection for internal threads defects with high reliability.

The method proposed in this paper introduces innovative concepts and practical strategies in few-shot defect detection, effectively addressing such challenges. Looking forward, there are many practical applications for few-shot defect detection in machine vision, such as in construction, energy, and healthcare. The model generalization application of this method to solve such problems can be expected.

## Figures and Tables

**Figure 1 sensors-24-05636-f001:**
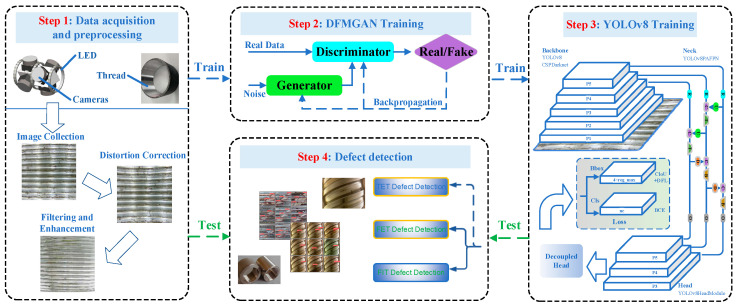
System framework flowchart.

**Figure 2 sensors-24-05636-f002:**
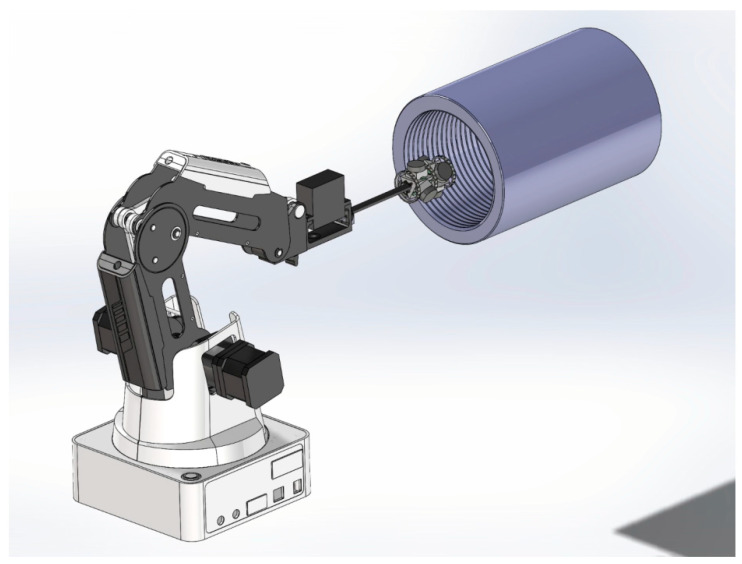
Establishment of an Internal Thread Image Acquisition System Model.

**Figure 3 sensors-24-05636-f003:**
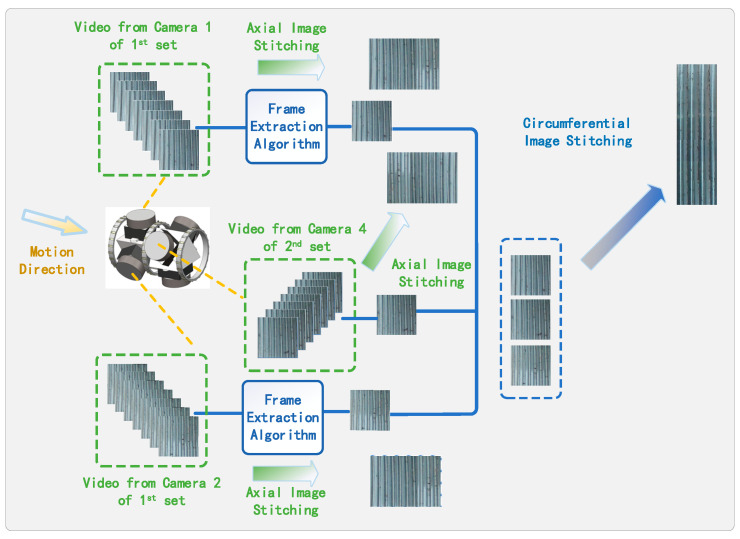
Feature image extraction algorithm based on video.

**Figure 4 sensors-24-05636-f004:**
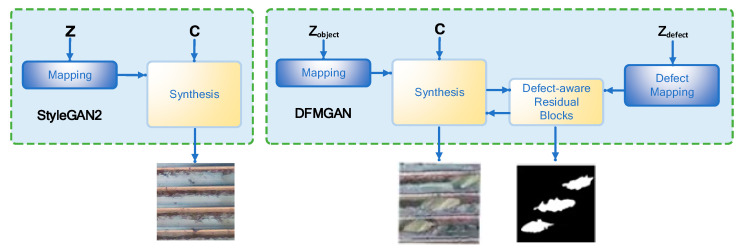
Internal thread defect image generation process based on DFMGAN.

**Figure 5 sensors-24-05636-f005:**
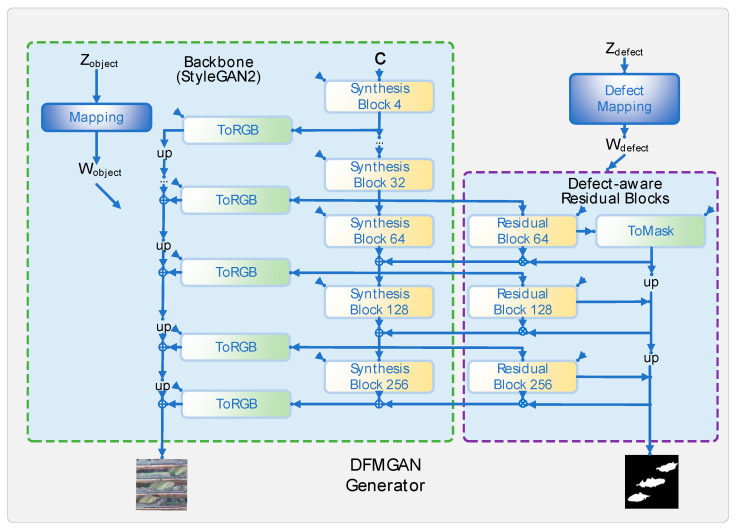
Structure of generator for a DFMGAN.

**Figure 6 sensors-24-05636-f006:**
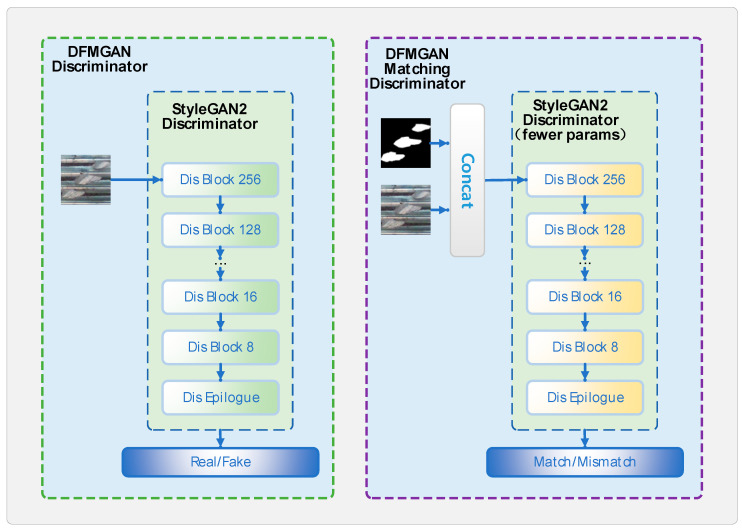
Structure of Discriminator for the DFMGAN.

**Figure 7 sensors-24-05636-f007:**
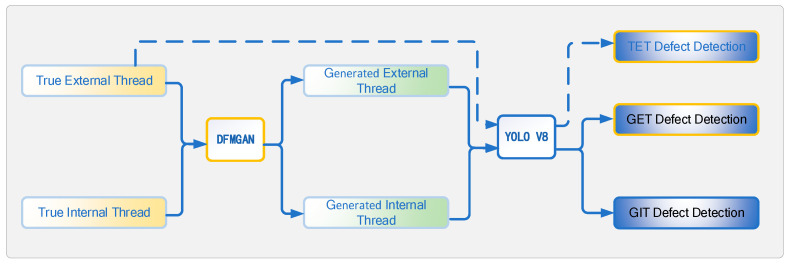
Process of defect image generation and detection.

**Figure 8 sensors-24-05636-f008:**
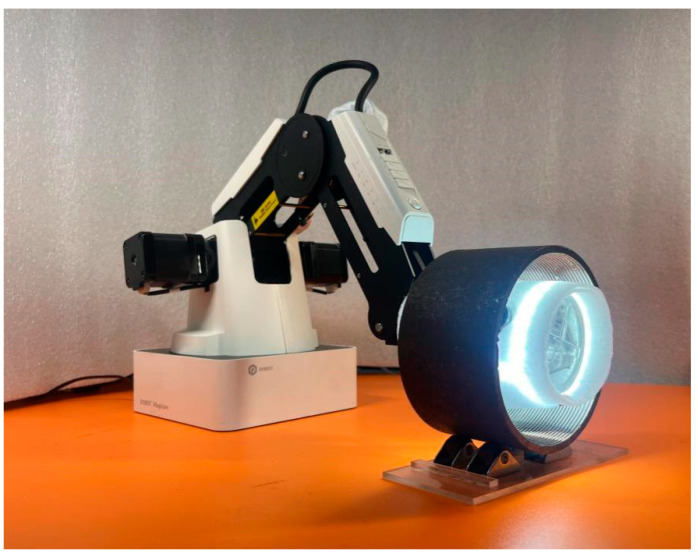
Internal thread image acquisition system.

**Figure 9 sensors-24-05636-f009:**
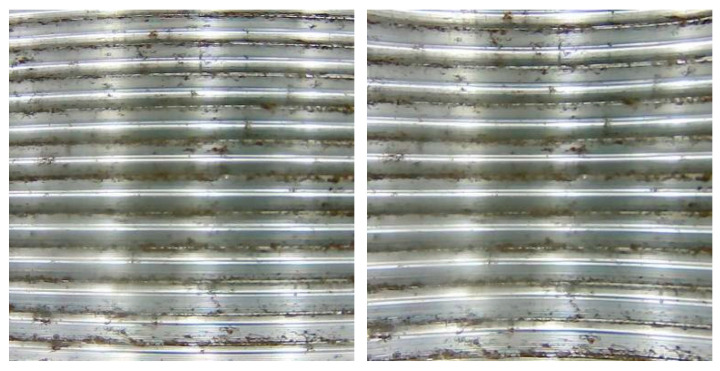
The original image (**left**) and the corrected image (**right**).

**Figure 10 sensors-24-05636-f010:**
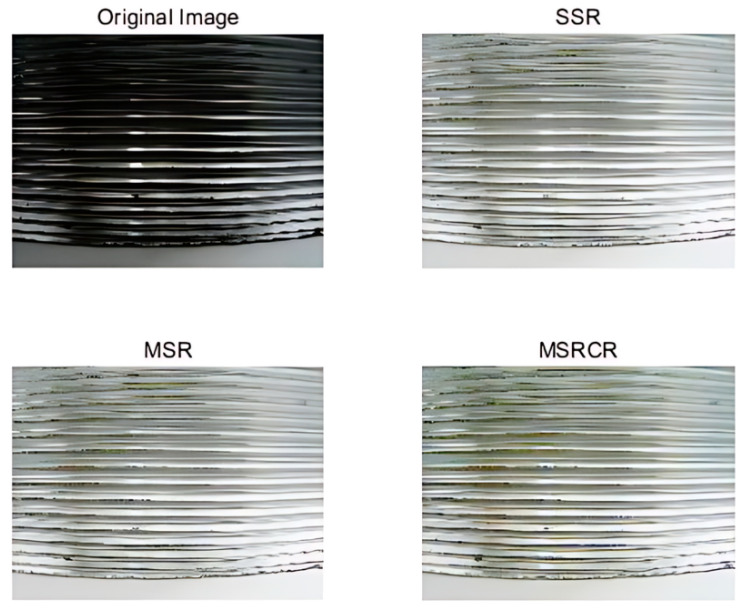
Comparison of Image enhancement effect.

**Figure 11 sensors-24-05636-f011:**
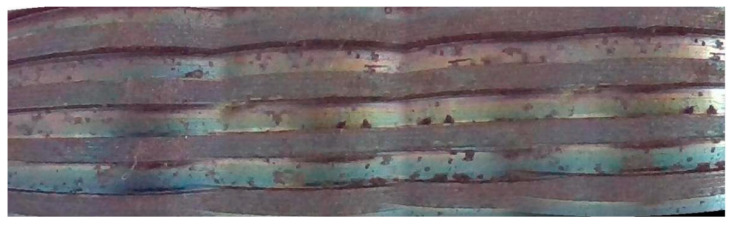
Stitching result of internal threads.

**Figure 12 sensors-24-05636-f012:**
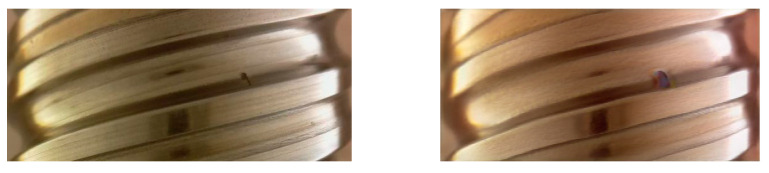
External thread real image (**left**) and generated image (**right**).

**Figure 13 sensors-24-05636-f013:**
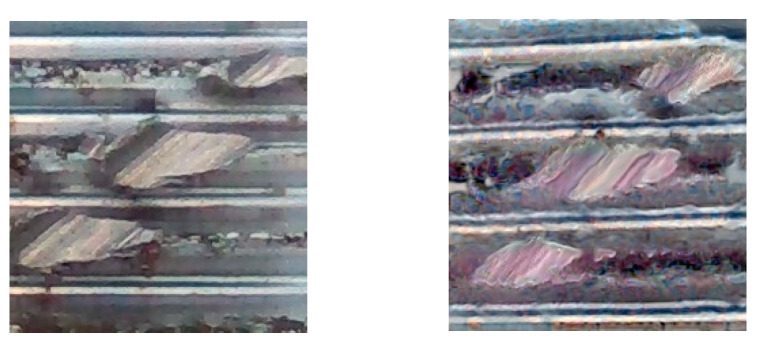
Internal thread real image (**left**) and generated image (**right**).

**Figure 14 sensors-24-05636-f014:**
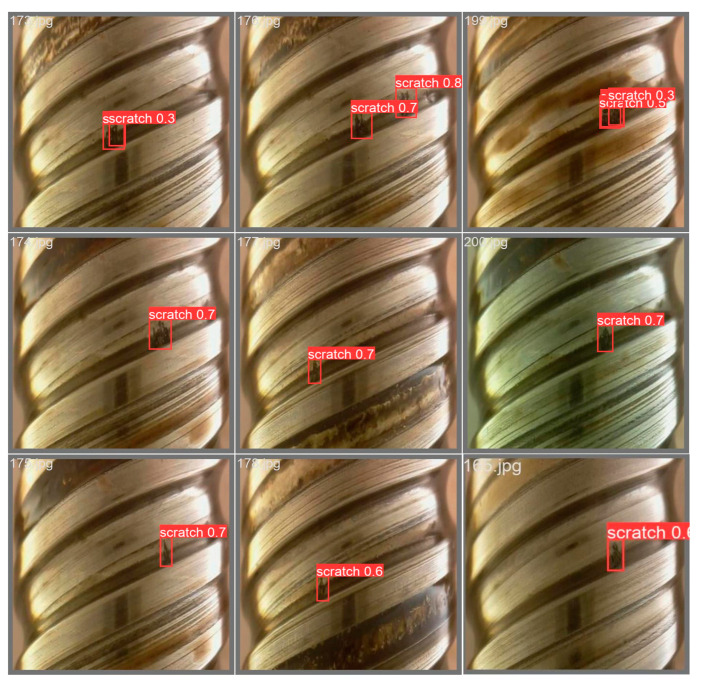
Scratch defect detection of external thread.

**Figure 15 sensors-24-05636-f015:**
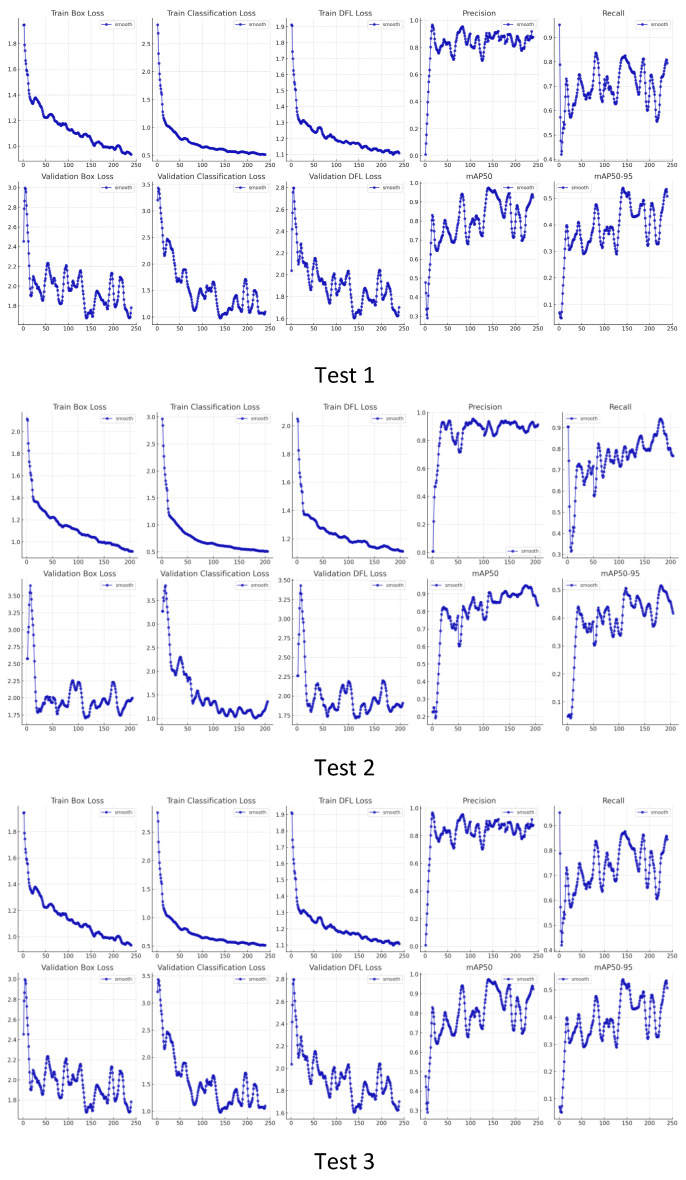
Process curves based on internal thread defect training.

**Figure 16 sensors-24-05636-f016:**
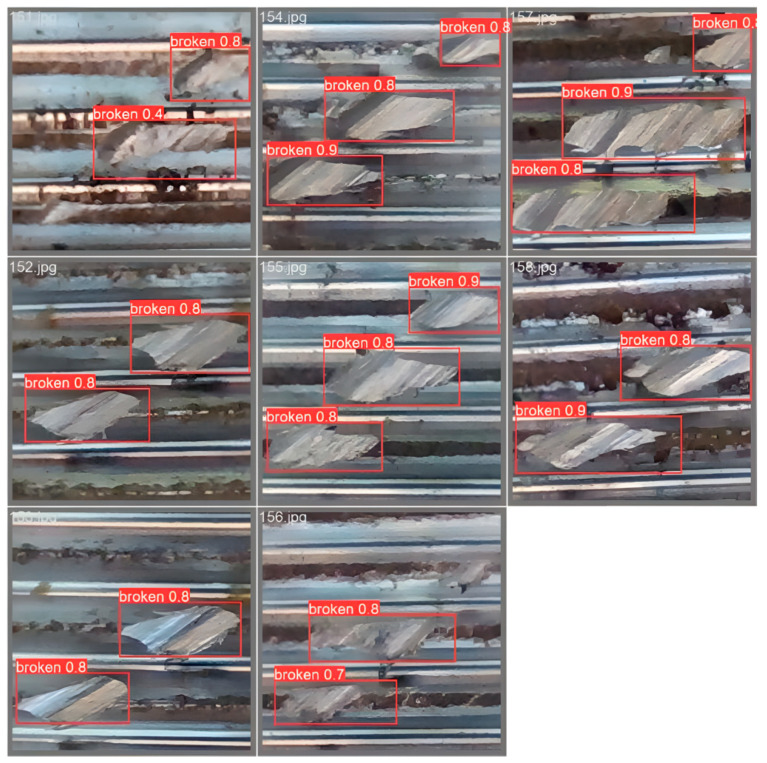
Real internal thread defect detection of the broken profile.

**Table 1 sensors-24-05636-t001:** The characteristics of evaluation metrics.

	Description	Suitability	Limitations
SSIM	Assesses perceived image quality by comparing luminance, contrast, and structure.	Ideal for evaluating visual quality by considering structural information.	Complex to compute and interpret.
UPMN	Indicates prediction confidence and potential variability.	Assess model reliability, not image quality.	Relevant to model uncertainty, not direct image quality measure.
WSMD	Evaluates differences in image parts using weighting similarity and Manhattan distance.	Useful where image parts have varying importance.	Does not reflect human visual perception directly.
PSNR	Measures signal-to-noise ratio in decibels.	Common for image reconstruction quality.	Focuses on pixel-level differences, not perceptual quality.

**Table 2 sensors-24-05636-t002:** Comparison of Filter Algorithms.

Feature	Non-Local Means	Median	Mean	3D Block-Matching	DnCNN
PSNR (FILTER)	26.10 dB	30.06 dB	30.74 dB	34.49 dB	32.7652 dB
SSIM (FILTER)	0.6805	0.8389	0.8738	0.9403	0.9058

**Table 3 sensors-24-05636-t003:** The results based on SSIMs and uncertainty scores.

	SSIM (GAN)↑	Uncertainty Score (GAN)↓
External thread	0.7933	0.3986
Internal thread	0.7371	0.4031

**Table 4 sensors-24-05636-t004:** Comparison of real and generated defect detection of external thread.

	Precision	Recall	mAP
External thread (real)	92.03%	90.70%	93.16%
External thread (generated)	93.92%	81.31%	84.68%

**Table 5 sensors-24-05636-t005:** The internal and external threads generate defect image training results.

	Precision (Mean ± SD)	Recall (Mean ± SD)	mAP (Mean ± SD)
Internal thread (generated)	94.27% ± 1.2%	79.53% ± 2.3%	88.25% ± 1.5%
External thread (generated)	93.92% ± 0.9%	81.31% ± 1.8%	84.68% ± 2.1%

## Data Availability

Data supporting the findings of this study are available from the corresponding author, Gengpei Zhang, upon reasonable request.

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
