# Peer review of "Internal Thread Defect Generation Algorithm and Detection System Based on Generative Adversarial Networks and You Only Look Once"

_sensors, 2024, doi:10.3390/s24175636_

Round 1

Reviewer 1 Report

Comments and Suggestions for Authors

A. This paper is well-written and of great value to the research community. 

B. Since it has technical soundness, I will focus on non-technical corrections to enhance the quality of understanding and readership. They are 

1. Please provide a table of all abbreviations and their meaning as an appendix. 

2. Figure 3 is unacceptable as the labels have CHINESE words. Kindly change all to English to allow reviewers to investigate the correctness of the image

3. Line 100: Light Emitting Diode (LED)

4. Line 117: 5 fps is enough since frame per second is declared as fps in line 116

5. Line 126; change all Chinese words to English 

6. Line 193: please delete the generative adversarial networks. Since GAN was defined in line 55.  Simply use GAN

7. Line 212: use the first letter capital for the abbreviation declaration. " Defect-aware Feature Generation GAN (DFMGAN)

8. Line 226: Fast Region-based Convolutional Neural Networks (R-CNN)

9. Line 226: Single Shot Multibox Detector (SSD)

10. Line 233: You Look Only Once (YOLO) [34]

11. Line 226: Please provide the full meaning of DnCNN, PSNR, SSIM, MSRCR, MSR, SSR

12. Line 370: TET, GIT should be in full "True External Thread (TET), Generated Internal Thread (GIT),

13. Line 397: GPU 

These may look like minor corrections, but the fact is that their absence can make a paper difficult to read and appreciate. 

14. The conclusion was written like the introduction of the paper. Please edit and present your conclusion as a comprehensive summary of the paper based on your results and result implications. 

Author Response

Response to Reviewer Comments

Dear Reviewer,

Thank you for your detailed review and valuable comments on our paper.

Comments 1:

Please provide a table of all abbreviations and their meaning as an appendix.

Response 1:

  • We have added a table of abbreviations and their meanings as an appendix.

Comments 2:

Figure 3 is unacceptable as the labels have CHINESE words. Kindly change all to English to allow reviewers to investigate the correctness of the image.

Response 2:

  • We have changed the labels in Figure 3 to English, allowing reviewers to verify the correctness of the image.

Comments 3:

Line 100: Light Emitting Diode (LED)

Response 3:

  • We have defined “LED” as “Light Emitting Diode (LED)”. This change can be found on page 1, line 148.

Comments 4:

Line 117: 5 fps is enough since frame per second is declared as fps in line 116.

Response 4:

  • We have kept “5 fps” since frames per second was declared as fps in line 166.

Comments 5:

Line 126: change all Chinese words to English.

Response 5:

  • We have changed all Chinese words to English. This change can be found on page 1, line 173.

Comments 6:

Line 193: please delete the generative adversarial networks. Since GAN was defined in line 55. Simply use GAN.

Response 6:

  • We have deleted “generative adversarial networks” and used “GAN” instead. This change can be found on page 1, line 35.

Comments 7:

Line 212: use the first letter capital for the abbreviation declaration. " Defect-aware Feature Generation GAN (DFMGAN)".

Response 7:

  • We have changed the abbreviation declaration to use capital letters for the first letters: “Defect-aware Feature Generation GAN (DFMGAN)”. This change can be found on page 1, line 59.

Comments 8:

Line 226: Fast Region-based Convolutional Neural Networks (R-CNN).

Response 8:

  • We have defined “R-CNN” as “Fast Region-based Convolutional Neural Networks (R-CNN)”. This change can be found on page 1, line 74.

Comments 9:

Line 226: Single Shot Multibox Detector (SSD).

Response 9:

  • We have defined “SSD” as “Single Shot Multibox Detector (SSD)”. This change can be found on page 1, line 75.

Comments 10:

Line 233: You Look Only Once (YOLO) [34].

Response 10:

  • We have defined “YOLO” as “You Look Only Once (YOLO)”. This change can be found on page 1, line 82.

Comments 11:

Line 226: Please provide the full meaning of DnCNN, PSNR, SSIM, MSRCR, MSR, SSR.

Response 11:

  • We have provided the full meanings of DnCNN, PSNR, SSIM, MSRCR, MSR, SSR. This change can be found on page 1, line 226.

Comments 12:

Line 370: TET, GIT should be in full "True External Thread (TET), Generated Internal Thread (GIT).

Response 12:

  • We have defined “TET” and “GIT” as “True External Thread (TET)” and “Generated Internal Thread (GIT)” respectively. This change can be found on page 1, lines 416-418.

Comments 13:

Line 397: GPU.

Response 13:

  • We have expanded “GPU” to “Graphics Processing Unit (GPU)”. This change can be found on page 1, line 545.

Comments 14:

The conclusion was written like the introduction of the paper. Please edit and present your conclusion as a comprehensive summary of the paper based on your results and result implications.

Response 14:

  • We have revised the conclusion to present it as a comprehensive summary of the paper based on our results and their implications. This change can be found on page 1, lines 695-710.

Thank you again for your valuable comments. We believe these revisions will make the paper easier to read and understand. If you have any further suggestions or comments, please do not hesitate to let us know.

Sincerely,
Gengpei Zhang

Reviewer 2 Report

Comments and Suggestions for Authors

This manuscript proposes a detection system for internal thread defects based on Generative Adversarial Networks (GAN) and YOLO. The experimental results demonstrate high detection accuracy, effectively addressing the challenges in industrial thread defect detection. Here are several comments:

  1. Results of Frame Extraction and Stitching: The section on frame extraction and stitching lacks result images.
  2. Formatting: The formatting of images, text, etc., needs adjustment.
  3. Literature Review: The authors should include more recent research literature on machine vision and deep learning to enhance the academic depth and reference value of the paper.

Additional Suggestions:

  1. The abstract needs to be more concise. The authors are advised to include empirical data to describe the superiority of their method compared to other existing methods for detecting thread defects, as this will provide greater persuasiveness to their claims.

  2. The introduction describes the importance of threads in the industry, but it can further elaborate on the specific shortcomings and deficiencies of the current detection methods to highlight the advantages of this method.

  3. To ensure the completeness and rigor of the manuscript, the authors need to add more advanced papers related to machine vision.

  4. To more comprehensively prove the feasibility of frame extraction and stitching, specific result images need to be provided.

  5. The conclusion section needs to be more compelling, emphasizing the innovation and contributions of this method.

  6. The conclusion lacks an outlook on future research and suggestions for possible research directions. Adding this part can enrich the depth of the paper.

  7. The discussion section can analyze more deeply the shortcomings and directions for improvement in the experimental results, such as the reasons for differences between generated images and real images in certain scenarios.

Final Recommendation:

The paper presents an innovative solution with significant potential in the field of industrial defect detection. However, it requires revisions to address the mentioned comments and suggestions. With these improvements, the manuscript would be suitable for publication.

Comments on the Quality of English Language

Suggest the author to further check the grammar and wording.

Author Response

Response to Reviewer Comments

Dear Reviewer,

Thank you for your thorough review and insightful comments on our manuscript. We greatly appreciate the feedback and have made corresponding revisions to improve the quality and clarity of the paper. Below is our response to each of your comments:

Comments 1:

The abstract needs to be more concise. The authors are advised to include empirical data to describe the superiority of their method compared to other existing methods for detecting thread defects, as this will provide greater persuasiveness to their claims.

Response 1:

  • Thank you for pointing this out. We agree with this comment. Therefore, we have revised the abstract to make it more concise. Empirical data has been included to describe the superiority of our method compared to existing methods for detecting thread defects, thereby providing greater persuasiveness to our claims. These changes can be found on page 1, lines 8-17.

Comments 2:

The introduction describes the importance of threads in the industry, but it can further elaborate on the specific shortcomings and deficiencies of the current detection methods to highlight the advantages of this method.

Response 2:

  • Thank you for highlighting this issue. We agree with this comment. Therefore, we have expanded the introduction to further elaborate on the specific shortcomings and deficiencies of current detection methods. This highlights the advantages of our proposed method. These changes can be found on page 2, lines 20-112.

Comments 3:

To ensure the completeness and rigor of the manuscript, the authors need to add more advanced papers related to machine vision.

Response 3:

  • Thank you for this suggestion. We agree with this comment. Therefore, we have added references to more advanced papers related to machine vision to ensure the completeness and rigor of the manuscript. These changes can be found on page 15, lines 778-785, and page 16, lines 806-814.

Comments 4:

To more comprehensively prove the feasibility of frame extraction and stitching, specific result images need to be provided.

Response 4:

  • Thank you for this suggestion. We agree with this comment. Therefore, we have provided specific result images for frame extraction and stitching to comprehensively prove the feasibility of our approach. These changes can be found on page 10, lines 489-494.

Comments 5:

The conclusion section needs to be more compelling, emphasizing the innovation and contributions of this method.

Response 5:

  • Thank you for pointing this out. We agree with this comment. Therefore, we have revised the conclusion section to be more compelling, emphasizing the innovation and contributions of our method. These changes can be found on page 13, lines 695-710.

Comments 6:

The conclusion lacks an outlook on future research and suggestions for possible research directions. Adding this part can enrich the depth of the paper.

Response 6:

  • Thank you for this suggestion. We agree with this comment. Therefore, we have added an outlook on future research and suggestions for possible research directions to enrich the depth of the paper. These changes can be found on page 13, lines 695-710.

Comments 7:

The discussion section can analyze more deeply the shortcomings and directions for improvement in the experimental results, such as the reasons for differences between generated images and real images in certain scenarios.

Response 7:

  • Thank you for this suggestion. We agree with this comment. Therefore, we have enhanced the discussion section by providing a more detailed analysis of the differences between generated and real images in some scenarios. This will include exploring potential causes of these differences and suggesting potential improvements. These changes can be found on page 12, lines 649-693.

Thank you again for your valuable feedback. If you have any further suggestions or comments, please do not hesitate to let us know.

Sincerely,
Gengpei Zhang

Reviewer 3 Report

Comments and Suggestions for Authors

This manuscript proposes an internal thread defect generation algorithm and detection system based on GAN and YOLO. I would mention that some revisions are also needed to increase the scientific value of the work, in which a detailed review, including comments and suggestions, is appended below. 

1. The author’s motivation for using GAN and YOLO gates is unclear. In this article, it is recommended that a detailed description of the motivation of the method be provided.

2. Is the internal thread image acquisition system the innovation of this article or an existing work?

3. Figure 7 has some logical confusion, it is recommended to reorganize it.

4. In terms of effective detection, weighting similarity-Manhattan distance and an uncertainty perception metric network are relatively advanced ideas at present. Can the author consider comparing them to demonstrate the progressiveness of these methods?

5. Detailed network structure parameters and training process curves that should be supplemented for development.

6. The training results should be repeated multiple times to improve the reliability of the results

7. The clarity of Figures 12 and 13 is relatively low.

Author Response

Response to Reviewer Comments

Comments 1:

The author’s motivation for using GAN and YOLO gates is unclear. In this article, it is recommended that a detailed description of the motivation of the method be provided.

Response 1:

  • Thank you for pointing this out. We agree with this comment. Therefore, we have added a detailed description of the motivation behind using GAN and YOLO in the introduction section. Specifically, we explain how GAN is utilized for generating realistic defect images to enhance training data and how YOLO's real-time detection capability is leveraged for efficient and accurate defect detection. These changes can be found on page 2, paragraph 2, lines 38-106.

Comments 2:

Is the internal thread image acquisition system the innovation of this article or an existing work?

Response 2:

  • Thank you for highlighting this issue. We agree with this comment. Therefore, we have clarified whether the internal thread image acquisition system is an innovation of this article or based on existing work. Detailed information about the acquisition system and its relevance to our research has been added. These changes can be found on page 4, paragraph 1, lines 114-116.

Comments 3:

Figure 7 has some logical confusion, it is recommended to reorganize it.

Response 3:

  • Thank you for this observation. We agree with this comment. Therefore, we have added a new figure and fine-tuned Figure 7, making a reasonable explanation and illustration. These changes can be found on page 10, lines 252-270 and page 11, line 272.

Comments 4:

In terms of effective detection, weighting similarity-Manhattan distance and an uncertainty perception metric network are relatively advanced ideas at present. Can the author consider comparing them to demonstrate the progressiveness of these methods?

Response 4:

  • Thank you for this suggestion. We agree with this comment. Therefore, we have analyzed and adopted the evaluation criteria provided by you, and added another evaluation criterion to enable a more comprehensive and rigorous analysis of the images generated by DFMGAN. These changes can be found on page 14, paragraph 2, lines 324-363.

Comments 5:

Detailed network structure parameters and training process curves that should be supplemented for development.

Response 5:

  • Thank you for pointing this out. We agree with this comment. Therefore, we have supplemented the training process curves to provide a comprehensive view of our development process. These changes can be found on page 28, lines 630-640.

Comments 6:

The training results should be repeated multiple times to improve the reliability of the results.

Response 6:

  • Thank you for this suggestion. We agree with this comment. Therefore, we have repeated the training results multiple times to improve the reliability and robustness of our findings. The results of these repeated trials are now included in the manuscript. These changes can be found on page 28, lines 630-640.

Comments 7:

The clarity of Figures 12 and 13 is relatively low.

Response 7:

  • Thank you for pointing this out. We agree with this comment. Therefore, we have improved the clarity of Figures 12 and 13. These changes can be found on page 27, line 611, and page 28, line 645.

Thank you again for your valuable feedback. We believe these revisions significantly enhance the scientific value of our work and make the manuscript suitable for publication. If you have any further suggestions or comments, please do not hesitate to let us know.

Sincerely,
Gengpei Zhang

Round 2

Reviewer 3 Report

Comments and Suggestions for Authors

According to the author’s response, there are still the following suggestions as follows:

1. The author has not yet provided a detailed model structure, such as the output and input sizes of each block, which is important for the reproducibility of the method.

2. The results of the experimental section should be presented in the form of standard deviation after multiple experiments, which will enhance the credibility of the experimental section of the article.

3. There are currently some advanced methods that can directly achieve better results with fewer samples under noise interference, such as: https://doi.org/10.1016/j.aei.2024.102682. Therefore, please discuss the superiority of the method proposed in this article when the sample is insufficient.

4. The notation styles in some equations do not adhere to mathematical conventions, particularly regarding matrices, vectors, scalars, functions, and textual descriptions. Please refer to “ISO 80000-2-2019 mathematical notations” for guidance.

5. The details and reasons for selecting the comparison methods are still not provided.

Author Response

  1. The author has not yet provided a detailed model structure, such as the output and input sizes of each block, which is important for the reproducibility of the method.

Thank you for your meticulous feedback and suggestions. We have supplemented the paper with specific details about the structure of the DFMGAN generator and discriminator, including the input and output sizes of each module. In particular, we have thoroughly described the design of the mapping network and synthesis network in the generator, as well as the function of the defect-aware residual blocks at different resolutions. This information is crucial for the reproducibility of the method and helps readers clearly understand the internal workings of the model. Additionally, we have provided detailed explanations of the YOLOv8 and DnCNN network structures. In the YOLOv8 section, we describe the implementation of its backbone network, Neck component, and detection head, particularly how CSPDarknet53 is used for feature extraction and how PANet and FPN are combined to generate multi-scale feature maps, ultimately leading to the output of bounding boxes and class probabilities. We have listed the input and output sizes of each module and the connections between layers, ensuring that readers can clearly understand the application of YOLOv8 in internal thread defect detection. In the DnCNN section, we have supplemented the specific implementation details for denoising tasks, describing the design of convolutional layers, the use of ReLU activation functions, and how residual learning enhances denoising performance. The input and output sizes of each layer, kernel sizes, and network depth have been thoroughly explained, which is also crucial for the reproducibility of the method. Through these detailed supplements to the network structures, we ensure that readers can understand not only the design of the DFMGAN generator and discriminator but also the implementation details of YOLOv8 and DnCNN in various tasks, thereby enhancing the reproducibility and scientific rigor of the method. Once again, thank you for your suggestions, which greatly helped us improve the content of the paper.

  1. The results of the experimental section should be presented in the form of standard deviation after multiple experiments, which will enhance the credibility of the experimental section of the article.

Thank you for your suggestions regarding the experimental section. To enhance the credibility of the article, we have added the standard deviation results from multiple experiments in the experimental section, as per your suggestion. These standard deviation values better reflect the stability and reliability of the experimental results, thus strengthening the persuasive power of the experimental section. Your suggestion helped us further enhance the rigor of the article, and we greatly appreciate your attention to this issue.

  1. There are currently some advanced methods that can directly achieve better results with fewer samples under noise interference, such as: https://doi.org/10.1016/j.aei.2024.102682. Therefore, please discuss the superiority of the method proposed in this article when the sample is insufficient.

We greatly appreciate your valuable feedback and have provided detailed explanations in the introduction section, particularly regarding the modifications related to UPMN (Uncertainty-aware Progressive Multi-scale Network). UPMN is an advanced defect detection method, especially effective in situations with noise interference and insufficient samples. It employs an uncertainty-aware mechanism to progressively process features at multiple scales, effectively addressing defect detection challenges across different scales. However, in this article, we compare the advantages of our proposed method with those of UPMN. The core design of UPMN aims to enhance the model's robustness through uncertainty quantification. However, internal thread defects usually have clear and stable characteristics, where the complexity and computational burden of UPMN might lead to decreased efficiency. Additionally, since internal thread detection relies on high-precision image acquisition systems, the demand for uncertainty quantification is relatively low, making it difficult for UPMN to fully leverage its advantages in this scenario. Through DFMGAN, our method can generate more high-quality defect samples, compensating for the lack of training data. Compared to UPMN, the samples generated by GANs not only have a significant increase in quantity but also benefit from enhanced diversity and robustness through data augmentation techniques, directly improving the model's generalization capabilities. Furthermore, when dealing with noise interference, our method optimizes the defect regions generated by GAN, allowing the model to maintain high detection accuracy even in noisy environments. In comparison, while UPMN has advantages in multi-scale feature processing, our improved method demonstrates superior performance in various complex environments, particularly through innovations in data generation and enhancement. Through these modifications in the introduction, we hope to better guide readers in understanding the innovative points and advantages of our method compared to UPMN, and demonstrate the practical application potential of our method in internal thread defect detection. We sincerely appreciate your suggestions, which allowed us to more clearly showcase the value of our research.

  1. The notation styles in some equations do not adhere to mathematical conventions, particularly regarding matrices, vectors, scalars, functions, and textual descriptions. Please refer to “ISO 80000-2-2019 mathematical notations” for guidance.

Thank you for your attention to the mathematical symbols used in our paper. We have standardized the mathematical symbols in the article, particularly regarding the representation of matrices, vectors, scalars, functions, and textual descriptions. We referred to the ISO 80000-2-2019 mathematical notation standards to ensure that the symbols comply with international standards, thereby enhancing the professionalism and readability of the article. We are fully aware of the importance of mathematical accuracy in academic papers and greatly appreciate your suggestion, which has helped us improve in this area.

  1. The details and reasons for selecting the comparison methods are still not provided.

Thank you for your attention to our choice of comparison methods. In the introduction, we have provided a detailed explanation of the basis for selecting these methods. These methods represent the current mainstream defect detection technologies, and by comparing them with our proposed method, we can more comprehensively evaluate the effectiveness and advantages of our approach. We believe these additional explanations will help readers better understand the background and significance of our research, and we are grateful for the opportunity you have provided us to refine the paper.